# Combining adaptive algorithms and hyper-gradient method: A performance and robustness study

## Abstract

Wilson et al. (2017) showed that, when the stepsize schedule is properly designed, stochastic gradient generalizes better than ADAM (Kingma & Ba, 2014). In light of recent work on hypergradient methods (Baydin et al., 2018), we revisit these claims to see if such methods close the gap between the most popular optimizers. As a byproduct, we analyze the true benefit of these hypergradient methods compared to more classical schedules, such as the fixed decay of Wilson et al. (2017). In particular, we observe they are of marginal help since their performance varies significantly when tuning their hyperparameters. Finally, as robustness is a critical quality of an optimizer, we provide a sensitivity analysis of these gradient based optimizers to assess how challenging their tuning is.

## 1 Introduction

Many new algorithms have been proposed in recent years for the minimization of unconstrained nonconvex functions such as the loss used in deep neural networks. A critical parameter of all these methods is the stepsize. A poor choice for that stepsize can either lead to very slow training or divergence. Worse, in many cases, the stepsize leading to the fastest minimization is the largest one achieving convergence, making the search difficult.

The dramatic effect of that tuning motivated the development of a range of optimization methods trying to integrate temporal metrics to adapt the stepsize for each parameter during optimization (Duchi et al., 2011; Zeiler, 2012; Tieleman & Hinton, 2012; Kingma & Ba, 2014). In particular, ADAM (Kingma & Ba, 2014) has become the default choice for many researchers and practitioners. One reason for this success is that these methods use a stepsize that is approximately normalized, that is the optimal stepsize potentially varies less across datasets and architectures than the optimal stepsize for non-adaptive methods.

However, Wilson et al. (2017) analyzed in depth the impact of adaptive methods on both training and generalization and showed that stochastic gradient methods with a carefully tuned stepsize could reach a lower generalization error than methods optimizing one stepsize per parameter. In order to achieve this result, a well tuned learning rate along with a suitable decaying schedule was required.

A recent approach for tuning the learning rate online was proposed by Baydin et al. (2018). This method, called *hypergradient descent* (HD), does not require setting a decay schedule ahead of time. One may thus wonder if, by automatically tuning the stepsize, such a technique would remove the last remaining advantage of adaptive methods, i.e. easier tuning. Our work relates this technique to the recent criticism made about the adaptive gradient methods (Wilson et al., 2017) and reconsider the value of these methods compared to their non-adaptive counterparts.

More precisely, this paper aims at extending the analysis of Wilson et al. (2017) in the following ways:

- How competitive is the recent online hypergradient scheme proposed by Baydin et al. (2018) compared to the offline scheme of Wilson et al. (2017)?

- Does this online scheme change the conclusions of Wilson et al. (2017)?

- Does this online scheme remove the need for fine-tuning the optimizer's hyperparameters, thereby removing the advantage of ADAM over stochastic gradient with momentum?
- What is the sensitivity of the learning rate schedule to a suboptimal choice of the hyperparameters?

The last point is often overlooked in the study of optimization methods. While investigating which training conditions bring the best performance led to significant progress in the field, the effort needed to have an optimizer perform at its best should be taken into account when evaluating the performance. Consider the following question: given a desired level of performance and limited computational ressources, which optimization method should be prefered and how should it be tuned? By this work, we would like to emphasize the value of tuning for gradient based methods, and what it can reveal about them.

## 2 BACKGROUND

We now review several techniques for stepsize selection. These techniques may either try to find a single global stepsize or one stepsize for each parameter in the model.

### 2.1 ADAPTIVE STEPSIZE

A first class of methods are the adaptive methods. To compute the parameter updates, these methods multiply the gradient with a matrix. When this matrix is diagonal, this is equivalent to using a distinctive stepsize per parameter. One of the first such method is Newton's method, multiplying the gradient with the inverse of the Hessian to minimize the local quadratic approximation to the loss. Similarly, Amari (1998) leveraged the Fisher information matrix to precondition the gradient. Later, Le Roux et al. (2008) used the covariance of the gradients to move less in directions of high uncertainty. A diagonal version of that technique, which adapts the learning rate on a per parameter basis, ADAGRAD, was proposed by Duchi et al. (2011). Subsequently, many works sought to improve upon ADAGRAD, such as RMSprop (Tieleman & Hinton, 2012), ADADELTA (Zeiler, 2012) and ADAM (Kingma & Ba, 2014). Other methods (Schaul et al., 2013) adapt the stepsize to minimize an estimate of the expected loss.

### 2.2 LEARNING RATE SCHEDULE

Despite learning a stepsize per parameter, adaptive methods still require a global stepsize, which might change during the course of optimization. We distinguish two classes of stepsize schedules: offline schedules, which are set ahead of time, and online schedules, which depend on the optimization trajectory.

OFFLINE LEARNING RATE SCHEDULES

Offline schedules define before the optimization, for each iteration $t$, the value of the stepsize $\alpha_t$. The seminal work of Robbins & Monro (1951) advocated for a schedule $\alpha_t = O(1/t)$, a rate often too slow in practice. Another decay schedule proposed by Bottou et al. (2018) is to halve the stepsize every time $t = 2^k$. The overall rate remains $O(1/t)$ but this method seems to work better in practice. A more aggressive schedule was proposed by Wilson et al. (2017) where the stepsize is halved every fixed number of iterations. While this leads to an exponential decay of the stepsize, precluding the method from reaching a local optimum from any initialization, this led them to the best results. In our work, we will call *lr-decay* the schedule where the stepsize is divided by 2 every 25 epochs.

ONLINE LEARNING RATE SCHEDULES

In contrast to offline schedules, online schedules tune the stepsize based on the dynamics of the optimization. Though extremely appealing, these methods have had little success, mostly because of their brittleness. Recently, however, Baydin et al. (2018) rediscovered the *hypergradient* update rule for the stepsize, which was originally suggested by Almeida et al. (1998). The optimisation of the stepsize can be framed as an iterative process that increases the stepsize when the last two gradients are aligned, suggesting that an acceleration is possible, and decreases it otherwise. It

requires no extra gradient computation and only needs a copy of the current gradient to be held in memory for the following iteration.

More precisely, assume we want to minimize a parametric function $f : \mathbb{R}^d \to \mathbb{R}$ with $d$ the number of parameters. Let us denote $\theta_t$ the parameter iterate at timestep $t$. A gradient descent update rule can be written:

$$\theta_t = \theta_{t-1} - \alpha \nabla_\theta f(\theta_{t-1}) = u(\Theta_{t-1}, \alpha)$$

where $\Theta_t = \{\theta_i\}_{i=0}^t$ and $\alpha$ is the learning rate. The goal is to move $\alpha_{t-1}$ toward the value that minimizes the expected objective function value at the next iteration, $\mathbb{E}[f(u(\Theta_t, \alpha))]$. Taking the derivative with respect to $\alpha$, we end up, under suitable assumptions, with the following learning rate update rule

$$\alpha_t = \alpha_{t-1} - \beta \tilde{\nabla}_\theta f(\theta_{t-1})^\top \nabla_\alpha u(\Theta_{t-2}, \alpha_{t-1})$$

where $\tilde{\nabla}_\theta f(\theta)$ is the noisy gradient of $f$. The authors observed that HD decreases the sensitivity to $\alpha_0$, the initial learning rate value, and that tuning only $\beta$ is sufficient. In the following section, we will see whether this lower sensitivity comes with the ability to recover nearly optimal optimization trajectory, or if it would still benefit from some tuning.

## 3 THE VALUE OF ONLINE LEARNING RATE ADAPTATION

Now, we focus on comparing *hypergradient* to the fixed decay scheme of Wilson et al. (2017). We show that, *i) hypergradient* can significantly benefit from a fine tuning of its hyperparameters, and by doing so, *ii)* it replicates, in most cases, the training performance of the best offline learing rate adaptation scheme and, that *iii)* stochastic gradient with *hypergradient* generalizes better than ADAM, extending Wilson et al. (2017) conclusions.

### 3.1 EXPERIMENTAL SETUP

We consider three gradient methods: SGD, SGD with Nesterov momentum (Sutskever et al., 2013) (SGDN) and ADAM. For each of these methods, we tried both *hypergradient* (SGD-HD, SGDN-HD, ADAM-HD) and the offline fixed decay (SGD-decay, SGDN-decay, ADAM-decay).

We study the performance of these six algorithms on three tasks: MNIST classification task using a feedforward neural network with two hidden layers of 1000 hidden units each, CIFAR-10 classification task using the VGG16 architecture from the Torch blog (Zagoruyko, 2015), and CIFAR-10 classification task using ResNet18 (He et al., 2016).

A grid search was performed to optimize the learning rate for each of these methods. The same grid of values was used for SGD and SGDN. For ADAM, another grid of comparable size was used. The scale of each grid was spread around the scale that is commonly known to be suitable for each of these optimizers. For the *hypergradient* method (HD), the grid search was conducted over both the initial learning rate $\alpha_0$ and the *hypergradient* learning rate $\beta$. However, for the learning rate fixed-decay (*lr-decay*), we kept the values of the decay factor and the decay frequency used in Wilson et al. (2017), which they have already optimized. The list of hyperparameters tried may be found in the appendix.

For momentum methods, we used $\mu = 0.9$. For Adam, we used the default setting with $\beta_1 = 0.9, \beta_2 = 0.999$ and $\epsilon = 10^{-8}$. L2 regularization with a coefficient of $10^{-4}$ was used for all our experiments.

Each experiement was conducted five times, using the same random initialization for all optimizers each time. CIFAR-10 was trained for 250 epochs and MNIST for 200 epochs.

We now present our results.

### 3.2 *Hypergradient* AND THE NEED OF TUNING

Before comparing *hypergradient* and *lr-decay*, we give a sense of how much *hypergradient* can boost the optimizers we are considering. We will later compare the best performing configurations to *lr-decay*.

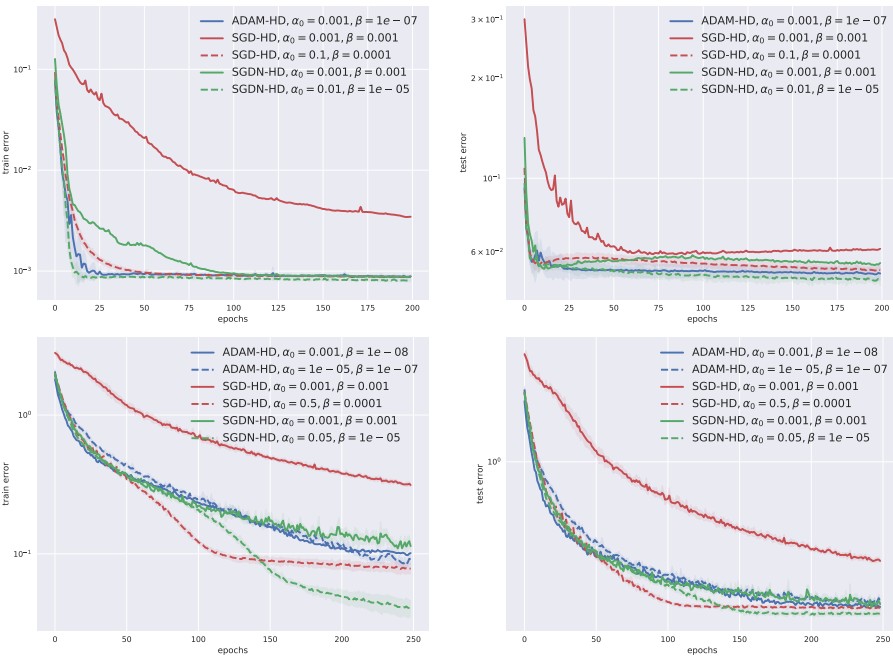

Figure 1: Comparison of the cross-entropy loss obtained with Hypergrad with the parameters suggested by Baydin et al. (2018) (solid line) and the optimized ones (dashed line) for a feedforward neural network on MNIST (*top*) and VGG16 on CIFAR-10 (*bottom*), both on the training (*left*) and the validation (*right*) sets.

Baydin et al. (2018) claim that *hypergradient* "*significantly reduces the need of the manual tuning of the initial learning rate for these commonly used algorithms*". For an arbitraty initial learning rate, *hypergradient* "*consistently brings the loss trajectory closer to the optimal one*". As can be seen in Figure 1, a slightly better tuning improves drastically the performance. Thus, even if *hypergradient* decreases the sensitivity to the inital learning rate for a fixed value of $\beta$, tuning both parameters remains essential.

Figure 1 gives the learning curves of *hypergradient* optimizers with tuned $\alpha_0$ and $\beta$. Compared to the results of Baydin et al. (2018), we now see that SGD-HD and SGDN-HD are competitive and able to outperform ADAM-HD. Our tuned configuration for ADAM-HD actually coincindes with the one given in Baydin et al. (2018).

### 3.3 ONLINE VS. OFFLINE LEARNING RATE ADAPTATION

After having optimized *hypergradient*, we now compare its best configuration to *lr-decay* as optimized in Wilson et al. (2017).

***Hypergradient* can replicate a close training perfomance to *lr-decay*.**    Figure 2 shows the training loss of *hypergradient* and *lr-decay* for the conducted experiments. When training on CIFAR-10, we observe that SGD-decay and SGDN-decay provide consistantly the best performance. We note that *hypergradient* is outperformed by *lr-decay* for ResNet18, but it is still able to reach a comparable performance on VGG net. In 5 out of 9 settings, *hypergradient* performs comparably to *lr-decay*. In the 4 others, it performs significantly worse.

***Hypergradient* generalizes worse than *lr-decay*.** Figure 3 shows the test loss for the experiments of Figure 2. We observe that *hypergradient* has a worse generalization performance than *lr-decay*, except SGDN-HD that is the best generalizing method for VGG net.

It is interesting to see that, by using the optimization dynamics in an online fashion, one can recover the training performance of a carefully tuned decay schedule.

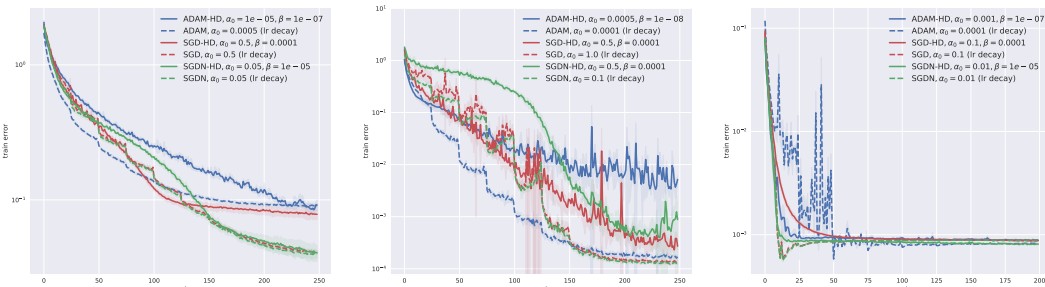

Figure 2: Train error of *lr-decay* and *hypergradient* for VGG16 on CIFAR-10 (*left*), Resnet18 on CIFAR-10 (*left*) and Feedforward neural network on MNIST (*left*). Shaded areas represent the 95% confidance interval around the mean performance over the 5 runs.

## 3.4 EXTENDING WILSON ET AL. (2017) CONCLUSIONS

From Figure 3, we can see, as in Wilson et al. (2017), that SGD-decay and SGDN-decay tend to generalize better than ADAM-decay for the VGG network. The same observation can be made for ResNet18. This conclusion is also valid for the *hypergradient* online scheme. Indeed, ADAM-HD is consistently outperformed by SGD-HD and SGDN-HD. So, *hypergradient* is also providing stochastic gradient with a learning rate schedule that helps it generalize better than ADAM. This recurrent pattern about adaptive gradient methods ability to generalize seems to suggest that they are less prone to benefit from learning rate decay schedules, whether offline or online; thus limiting their competitiveness.

This makes the *hypergradient* another learning rate adaptation technique that invites us to reconsider the use of ADAM, as an adaptive gradient algorithm, to train neural networks. The main practical message here is that it's preferable to combine the stochastic gradient with a well designed learning rate decay. Momentum is also adviced to complete this combination.

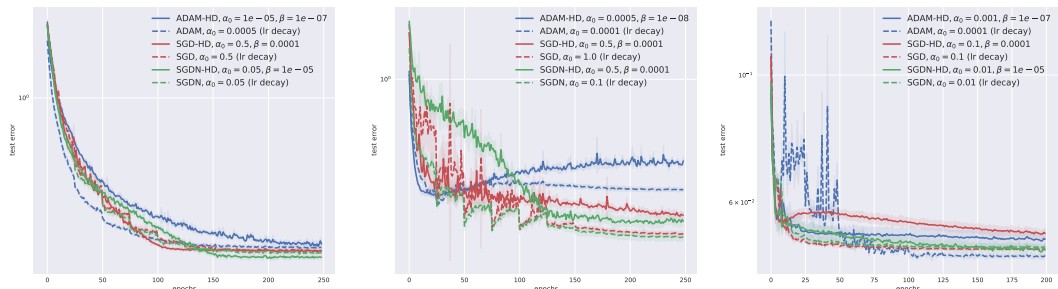

Figure 3: Test error of *lr-decay* and *hypergradient* for VGG16 on CIFAR-10 (*left*), Resnet18 on CIFAR-10 (*left*) and Feedforward neural network on MNIST (*left*). Shaded areas represent the 95% confidance interval around the mean performance over the 5 runs.

So far, we have seen what these gradients methods with online and offline learning rate schedules can achieve. Tuning their hyperparameters revealed the potential of some and limitations of others. In the next section, we give an analysis of the sensitivity of such tuning with respect to the initial stepsize and describe the effect of the described learning rate adaptation methods.

# 4 SENSITIVITY ANALYSIS OF GRADIENT METHODS

In order to better understand how difficult setting the right stepsize can be, we plot the performance of each optimizer as a function of its learning rate [1], where performance is defined as the minimal cross-entropy loss, averaged over the 5 runs, reached by each method. We show that ADAM with a constant stepsize is as difficult to tune as stochastic gradient. Interestingly, we found that, while *lr-decay* allows SGD and AGDN to generalize better than ADAM, it also makes the latter easier to tune than the former.

In this section, we focus our sensitivity analysis on the CIFAR-10 classification problem using VGG and ResNet18 architectures, since they pmake the differences in the behaviour of the optimizers we are studying explicit. In this section, SGD, SGDN and ADAM will be used to denote the constant stepsize version of these gradient methods.

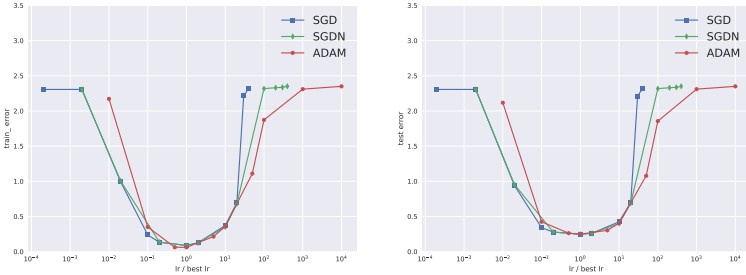

Figure 4: Sensitivity of gradient methods (constant stepsize) for VGG net to the learning rate, on CIFAR-10. On this architecture, these algorithms show a tight region around the best stepsize. Tuning them would require more carefulness.

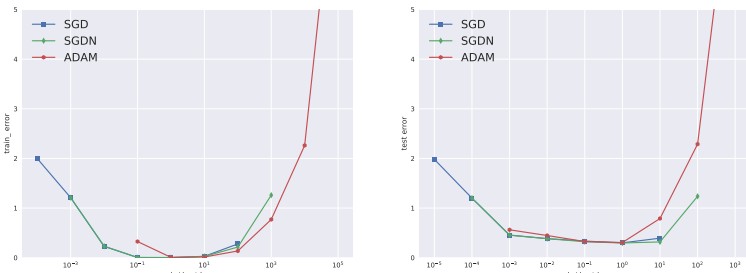

Figure 5: Sensitivity of gradient methods (constant step size) for ResNet18 to the learning rate, on CIFAR-10. The residual net offers a wider region around the best stepsize. The tuning them would require little effort.

To better compare the sensitivity of each optimizer in the neighbourhood of its best learning rate, we plotted the ratio to the best learning rate on the x-axis. Hence, the minimum value is always reached for a ratio of 1.

Figure 4 and 5 represent the training and test errors over the grid of inital learning rates for VGG net and ResNet18 respectively. On these figures, the gradient methods were used with a constant stepsize.

For VGG net, Figure 4 shows the performance of SGD and SGDN worsens faster when increasing the learning rate than when decreasing it. This observation suggest that one should aim at the highest step size that does not diverge, an observation which matches the theory on quadratic bowls. ADAM shows a similar U-shaped dependency of the performance with respect to the initial learning rate, but its behaviour is more symmetric.

In general, these observations raise the problem of how fine-grained should the set of learning rate values one is selecting from be. A wide valley means that fewer values are sufficient to find a well

---

[1] A similar figure was described by Rebecca Roelofs during their talk at NIPS'17.

performing configuration while a tighter one suggests that one requires to select from a finer range of values.

**The architecture influences the tuning.** In Figure 5, we see that all the three gradient methods exhibit a wider valley around the best learning rate value, which implies that it is easier to tune the learning rate for a ResNet18 than for a VGG16. This importance of the architecture choice in the tuning of the learning rate is often overlooked in analysis comparing such architectures.

## 4.1 THE EFFECT OF *lr-decay* ON THE SENSITIVITY

The *lr-decay* scheme improved the training and generalization performance of gradient methods. Here, so that to complete our understanding of such learning rate adaptation approach, we investigate the way it changes the tuning sensitivity.

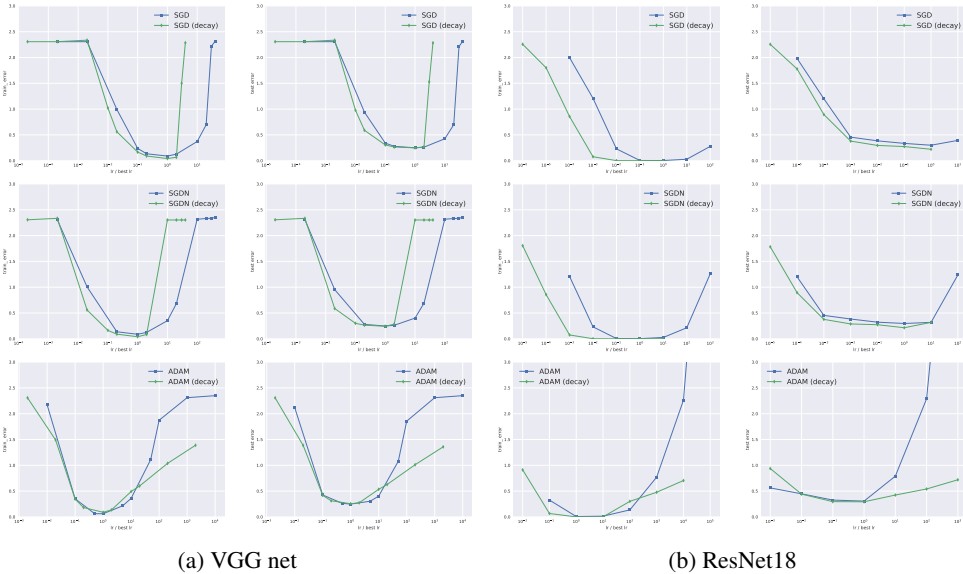

(a) VGG net                    (b) ResNet18

Figure 6: The effect of *lr-decay* on the learning rate sensitivity for VGG net and ResNet18 on CIFAR-10. *lr-decay* makes the tuning of ADAM easier. ResNet18 makes the gradient methods less sensitive to the stepsize choice.

Figures 6 shows the sensitivity of each gradient method (constant stepsize) along with its *lr-decay* version. For all these experiments, we can see that *lr-decay* moves the best learning rate value to a higher one (there are less values on the right of the best stepsize of *lr-decay*). Concerning the VGG architecture (and 6a), for SGD and SGDN, *lr-decay* improves over their performance with the best constant stepsize, but at the expense of a much sharper slope at the right of its own new best learning rate. In addition, for ADAM, *lr-decay* decreases the sentivity at the right of best stepsize which, makes tuning less challenging for ADAM-decay than SGD/SGDN-decay. Thus, the generalization potential that *lr-decay* brings to stochastic gradient (Wilson et al., 2017) comes at the expense of a trickier tuning. Moreover, for smaller scales of the learning rate *lr-decay* is not penalizing the performance. It even allows some improvements.

For ResNet18 (6b), *lr-decay* shows even more robustness to a suboptimal choice of the learning rate. The performance remains quite at the same level for a larger range of the learning rate scale. *lr-decay* seems to significantly decrease the sensitivity for ADAM, by flattening its U-shaped sensitivity curve, which confirms that ADAM-decay requires less tuning effort.

## 4.2 THE EFFECT OF *hypergradient* ON THE SENSITIVITY

Figure 7 shows the sensitivity of *hypergradient* methods to the stepsize $\alpha_0$, for different values of the hypergradient stepsize $\beta$, for a VGG net trained on the CIFAR-10 dataset.

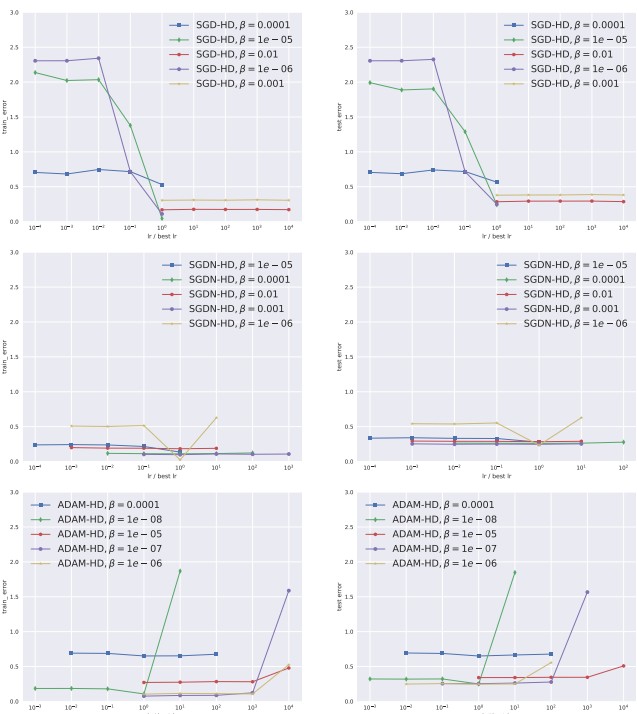

Figure 7: The effect of *hypergradient* on the learning rate sensitivity for VGG net and ResNet18 on CIFAR-10. For a given $\beta$ parameter, *Hypergradient* demonstrates some robustness to the initial learning rate. However to reach higher performance, tuning $\beta$ is crucial.

As claimed by Baydin et al. (2018), *hypergradient* methods have a low sensitivity to the choice of the stepsize $\alpha_0$, for a fixed hypergradient learning rate $\beta$. Indeed, on Figure 7, we see that, for several settings of $\beta$, the performance is nearly constant over the 5 stepsizes that we tried. We'd like to note, however, that some other $\beta$ settings demonstrate high sensitivity on some range of (the reduced) $\alpha_0$ value. For example, the train performance of SGD-HD with $\beta = 10^{-5}$ pis constant for stepsizes that are 100 times smaller than its best step size, but it drastically improves above that stepsize scale and outperforms all the other configurations.

An interesting point to add here, is that stochastic gradient with nesterov momentum (SGDN) happens to be, not only the gradient method that benefits the most from *hypergradient* in terms of performance and generalization (see section 3), but also the one that is less sensitive to the tuning compared to stochastic gradient and *Adam*.

## 5 CONCLUSION

We studied the impact of hypergradient methods on common optimizers and observed that it does not perform better than the fixed exponential decay proposed by Wilson et al. (2017). Further, while hypergradient is designed to simplify the tuning of the stepsize, it can still greatly benefit from a fine tuning of its hyperparameters. Finally, similar to the conclusions reached by Wilson et al. (2017), SGD and SGDN combined with a tuned hypergradient perform better than ADAM with the same method.

This study raises several questions. First, is it possible to derive an automatic stepsize tuner that works consistently well across datasets and architectures? Second, what would an optimizer tuned for robustness look like ? In any case, our results suggest that the current adaptive methods wouldn't be the best candidates to build on such an optimizer. One would rather augment the stochastic gradient with more promising learning rate schedules.

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

## A  THE LISTS OF STEPSIZES TRIED FOR TUNING

For SGD and SGDN, whether with constant stepsize, *lr-decay* or *hypergradient*, we used: $\{10^{-5}, 0.0001, 0.001, 0.005, 0.01, 0.05, 0.1, 0.5, 1\}$.

For SGD-HD and SGDN-HD, we selected $\beta$ from: $\{10^{-6}, 10^{-5}, 0.0001, 0.001, 0.01\}$.

For ADAM, whether with constant stepsize or *lr-decay*, we used: $\{10^{-6}, 10^{-5}, 5 \cdot 10^{-5}, 0.0001, 0.0005, 0.001, 0.005, 0.01, 0.1, 1\}$.

For ADAM-HD we selected $\alpha_0$ from: $\{10^{-6}, 10^{-5}, 0.0001, 0.0005, 0.001, 0.005, 0.01\}$ and $\beta$ from: $\{10^{-8}, 10^{-7}, 10^{-6}, 10^{-5}, 0.0001\}$.

