# OpenReview forum: "Combining adaptive algorithms and hypergradient method: a performance and robustness study"
_ICLR.cc/2019/Conference_

### Official Review · AnonReviewer2 · 2018-11-02
**An emperical study on several methods for adjusting learning rate**

**Rating:** 4
**Confidence:** 4

**Review:**

The paper reports the results of testing several stepsize adjustment related methods including  vanilla SGD, SGD with Neserov momentum, and ADAM. Also, it compares those methods with hypergradient and without. The paper reports several interesting results. For instance, they found hypergradient method on common optimizers doesn't perform better that the fixed exponential decay method propose by Wilson et al. (2017).

Though it is an interesting paper, but the main issue with this paper is that it lacks enough innovation with respect to theory or empirical study. It is not deep or extensive enough for publishing at a top conference.

On page 3, it will be better to explain why use mu = 0.9, beta, etc. Why use CIFAR-10, MNIST?

The URL in References looks out of bound.

---

### Official Review · AnonReviewer3 · 2018-11-07
**incremental empirical contribution**

**Rating:** 3
**Confidence:** 2

**Review:**

Clarity: Below average
- The introduction would be easier to follow if you named Baydin's approach and your own approach, because in the 2-4 bullet points you say "this online scheme", and "the learning rate schedule", without being perfectly clear what you are talking about
- The last sentence of the introduction is meant to clearly state your hypothesis, so I was expecting "emphasize the value of *", i.e. either adaptive or non-adaptive methods, rather than just general 'tuning', which is self-apparently important.

Quality: Below average
This is a purely empirical study that does not go too deep. It is not quite a review paper, but only compares previous methods.

Pros:
I especially appreciate the sensitivity analysis, ie Fig 6. If only all ML papers had something like this to suggest the difficulty of setting hyperparameters for their proposed methods.

Cons:
- You should use mathematics to describe what you are talking about with adaptive stepsize in Sec 2.1. "these methods multiply the gradient with a matrix". Just giving one equation would be extremely helpful.
- If I understand correctly, you are interpreting the inverse-Hessian as used in Newton's method and other non-diagonal 'gradient conditioners' as types of stepsize. This is definitely interesting, but again it would be very simple to see what you are saying with an equation instead of starting with the phrase "stepsize" which is generally understood to be a scalar multiple on the gradient.
- I'm surprised you jump right into experiements after your background settings. It's apparent that this paper fundamentally relies on the Wilson (2017) hypergradient paper. Your paper should be more self-contained: 'hypergradient' is not even defined in this paper, is it?...

Especially:
How do you know that if you change the model architecture, data, and loss, that a similar result will occur? I imagine that it heavily relies on the data and model-- in other words, that the sensitivity is dependent on "how an algorithm reacts to a certain data/loss/model landscape". I'm trying to say that I'm not convinced these results generalize to any other situation than the one presented here (so does it really say anything about the different stepsize selection rules?)

Random side note:
Since your appendix is only a few lines, you could consider succinctly listing learning rates with set notation, for example {1e-n,5e-n : -5<n<1}.

---

> ### Comment · AnonReviewer3 · 2018-11-07
> **my apologies**
>
> Sorry, I  meant to erase the comment "which is self-apparently important", which isn't appropriate and doesn't make sense.

---

### Official Review · AnonReviewer1 · 2018-11-07
**A technical report rather than a research paper**

**Rating:** 3
**Confidence:** 4

**Review:**

General:
In general, this looks like a technical report rather than a research paper to me. Most parts of the paper are about the empirical analysis of adaptive algorithms and hyper-gradient methods. The contribution of the paper itself is not sufficient to be accepted.

Possible Improvements:
1. The study of such optimization problem should consider incorporating mathematics analysis with necessary proof. e.g. show the convergence rate under specific constraints. Even the paper is based on others' work, the author(s) could have extended their work by giving stronger theory analysis or experiment results.
2. Since this is an experimental-based paper, besides CIFAR10 and MNIST data sets, the result would be more convincing if the experiments were also done on ImageNet(probably should also try deeper neural networks).
3. The sensitivity study is interesting but the experiment results are not very meaningful to me. It would be better if the author(s) gave a more detailed analysis.
4. The paper could be more consistent. i.e. emphasize the contribution of your own work and be more logical. I might miss something, but I feel quite confused about what is the main idea after reading the paper.

Conclusion:
I believe the paper has not reached the standard of ICLR. Although we need such paper to provide analysis towards existing methods, the paper itself is not strong enough.

---

### Author Response · Authors · 2018-11-08
**Thank you for your valuable comments**

We would like to thank the reviewers for their time. We will take their comments into account for a future version of this work.

---

### Public Comment · (anonymous) · 2018-11-11

On the positive side this paper performs several interesting experiments comparing various learning rate tuning algorithms.
The paper also spends time on sensitivity/robustness, which has not received adequate attention in the literature.
However, I am afraid there is no technical or methodological contribution from this paper that meets the ICLR standards.

Some feedback which will hopefully help in future submission:

1. Use clear definitions and notation to introduce methods. Currently, all methods are only described in words, and this creates confusion, especially for the "hypergradient method" that is new.

2. Be consistent about claims and results presented in the paper. For example, in the Abstract the claim is "We analyze the true benefit of these hypergradient methods..."  but not such analysis is presented. If your goal is experimentation and not analysis it is better to make that clear early.

3. As above, there is no "sensitivity analysis" offered in this paper. I do think this is an important subject and I applaud the authors for focusing on that. However, currently in the paper there are only experimental results and simulations.
There are limitations with the experiments as well. In Figures 4-5-6 we only get some plots on how the train error depends on the learning rate on some particular datasets. A deeper investigation would be helpful here as to why we see the results we see, so as to substantiate claims such as " Figure 4 shows the performance of SGD and SGDN worsens faster when increasing
the learning rate than when decreasing it." (page 6) It would be nice to get such general results, but this requires a deeper and more thorough investigation, whereas currently the evidence may be circumstantial.

4. Section 2.1 is a good place to start introducing notation. Although the referenced methods are known, it helps to lay out some notation so that readers have a clear idea what the authors have in mind.

5. Similar to point #1: p2 "but this method seems to work better in practice." Blanket statements are hard to accept without solid arguments. What does "better" mean here and what does "in practice" mean? Overall, the authors should avoid such statements without presenting solid evidence. Another example in p5: "It is interesting to see that, by using the optimization dynamics in an online fashion, one can recover the training performance of a carefully tuned decay schedule."

6. About sensitivity analysis the authors could also look into (Robust Implicit Backpropagation, Fagan & Iyengar, 2018) where the authors use implicit methods to stabilize fitting algorithms for neural networks. Could the ideas in that paper apply here?

---

> ### Author Response · Authors · 2018-11-12
> **Thanks for the constructive feedback**
>
> First, the authors would like to thank you for the time given to reviewing this paper and the constructive comments you are offering. We will take them into account for future submission.
>
> 1. , 4. , 5.  -  These are relevant suggestions and will be followed for the future version.
>
> 2. By "true benefit", we meant the gain in performance of Hypergradient when its hyperparameters are tuned more carefully (as we proved in the paper, hypergradient can significantly benefit from such tuning). However, we do agree this claim could be formulated in a more consistent way w.r.t to our results.
>
> 3. As in 2., the sensitivity analysis was only empirical. We will investigate a large set of experimental settings to support our observations.
>
> 6. Robust Implicit Backpropagation (Fagan & Iyengar, 2018)  is offering ideas that can perfectly fit in the landscape of this study. In fact, Implicit Backpropagation (IB) approximates the update of Implicit Stochastic Gradient Descent which is known to be stable and robust to learning rate. This makes it a good candidate to consider in an investigation like the one we are conducting, in order to check how IB compares to adaptive gradient methods and the various learning rate schedules we are considering. More specifically, IB seems to be very efficient for recurrent models. Since, we are planning to extend our investigation to tasks that correspond to recurrent models (e.g. language modelling), IB would definitely be a good method to compare to. Thank you again for sharing this reference. We will be considering it in a future version eventually.

---

### Meta-Review · Area_Chair1 · 2018-12-13

**Confidence:** 5
**Recommendation:** Reject

**Metareview:**

The paper is a premature submission that needs significant improvement in terms of conceptual, theoretical, and empirical aspects.